# Effect of Acute Intake of Fermented Orange Juice on Fasting and Postprandial Glucose Metabolism, Plasma Lipids and Antioxidant Status in Healthy Human

**DOI:** 10.3390/foods11091256

**Published:** 2022-04-27

**Authors:** Blanca Escudero-López, Isabel Cerrillo, Ángeles Ortega, Franz Martín, María-Soledad Fernández-Pachón

**Affiliations:** Área de Nutrición y Bromatología, Departamento de Biología Molecular e Ingeniería Bioquímica, Universidad Pablo de Olavide, Carretera de Utrera Km 1, E-41013 Sevilla, Spain; besclop@upo.es (B.E.-L.); icergar@upo.es (I.C.); maortega@upo.es (Á.O.); fmarber@upo.es (F.M.)

**Keywords:** fermented orange juice, acute intake, postprandial, lipids, glucose, antioxidant status, inflammation status, cardiovascular risk, functional food, healthy humans

## Abstract

Higher postprandial plasma glucose and lipemia, and oxidative and inflammatory responses, are considered important cardiovascular risk factors. Fermentation of fruits has generated products with high concentrations of bioactive compounds. The aim of this study was to evaluate the potential acute effects that fermented orange juice (FOJ) can exert in healthy humans by modulating postprandial response, and inflammatory/antioxidant status, compared with orange juice (OJ). Nine volunteers were recruited for a randomized, controlled, and crossover study. Participants ingested 500 mL of FOJ. At 4 h post intake, subjects consumed a standardized mixed meal. Blood samples were collected at 0–8 h hours post intake. The subjects repeated the protocol with OJ following a 2-week washout period. Glucose and lipid metabolism, plasma antioxidant capacity (ORAC, FRAP), endogenous antioxidants (albumin, bilirubin, uric acid), C-reactive protein and fibrinogen were measured in plasma samples. There was a trend of a smaller increase in LDL-C after FOJ intake compared with OJ, a significant decrease in apo-B and significant increase in ORAC. The glycemic and triglyceride response of meal was attenuated with FOJ. No differences were obtained in endogenous antioxidants and inflammation status between the treatments. The acute consumption of FOJ could play a protective role against cardiovascular risk factors.

## 1. Introduction

Chronic diseases are the main causes of mortality in industrialized countries. Among them, cardiovascular diseases (CVD) represent the first cause of death, followed by some types of cancers, chronic respiratory diseases and type II- diabetes [1,2]. The incidence and prevalence of some of these chronic diseases, such as atherosclerotic CVD and metabolic syndrome, has increased notably in recent years due, among other factors, to a dysregulation of the inflammatory response, increased oxidative stress and unbalanced lipid metabolism [3,4,5,6]. Lifestyle and eating patterns also play a key role in its development [7,8,9]. Experimental and epidemiological data show that regular consumption of fruits, vegetables and plant-derived beverages is associated with a lower risk of diet-related chronic diseases and metabolic impairments [10,11,12,13]. 

Among fruit juices, orange juice (OJ) is the most consumed worldwide and several intervention studies suggests that its regular consumption could provide additional protective activity against the onset and progression of several chronic diseases, such as CVD and some types of cancer [14,15]. These positive effects have been attributed to its profile of bioactive compounds such as vitamin C, (poly)phenols, carotenoids, folate, and melatonin [16,17,18] which exert a large number of biological activities: antioxidant activity, regulation of lipid profile, improvement of endothelial function, anti-inflammatory function, anti-cancer property, glycemic regulation, improvement of immune function, and reproductive and bone metabolism [19,20,21,22,23,24,25,26,27,28]. The healthy effects associated with the OJ consumption have promoted research in order to determine the most appropriate technological conditions to preserve or enhance its biological functions. Fermentation process conducted in fruit juices represents new products with higher concentration of phytochemicals and/or improved profile of bioactive compounds than the respective substrates [29,30,31,32]. In this sense, our group has described for the first time the profile of bioactive compounds of a beverage produced by controlled alcoholic fermentation of OJ, showing increases in flavanone, carotenoid and melatonin contents and an improvement in the antioxidant activity compared with the original juice [17,18,33,34]. The potential preventive and therapeutic role of fermented orange juice (FOJ) were previously shown in long-term studies, first with healthy mice and mice with metabolic disorder [35,36], and subsequently with humans. The data provided evidence that chronic consumption of FOJ (500 mL/day for 2 weeks) enhanced antioxidant status and decreased inflammation status and lipid peroxidation in healthy humans [37], and improved lipid profile and reduced endothelial dysfunction and oxidative stress in hypercholesterolemic subjects [38]. In addition, acute intake studies showed that FOJ can be considered a good dietary source of bioavailables (poly)phenol and β-cryptoxanthin [39,40,41].

On the other hand, high postprandial triglyceride and glucose plasmatic levels are accepted relevant risk factors in both dyslipidemic and diabetic patients and healthy subjects [42]. Recent studies have shown a strong relation between acute consumption of foods rich in bioactive compounds and improved postprandial lipemia, glucose metabolism and/or antioxidant and inflammatoty status [43,44,45]. 

The potential healthy effects of FOJ on postprandial metabolism (i.e., glucose and lipid postprandial responses) and antioxidant/inflammatory status have not been explored. We hypothesize that the healthy effects of FOJ on cardiovascular risk factors could be also mediated by its acute consumption. Most of the previous studies based on the acute effect of OJ or other fruit juices evaluated short periods of time (1–3 h), both in trials that included the joint intake of food (normal or high-fat or high-carbohydrate) or oral glucose with the fruit [46,47,48,49,50,51,52,53,54], and in trials that included the juice alone [45,55,56,57]. This study also presents two novel aspects to assess the potential effect of metabolites derived from bioactive compounds on postprandial glucose and triglyceride levels: the extension of the length of the measurement time up to 8 h, and the consumption of a food 4 h after ingesting the OJ or FOJ. In our previous bioavalilability studies, the maximum levels of carotenoids and (poly)phenol metabolites were obtained at 3 and 5 h post intake, respectively [40,41]. 

This study aimed to evaluate the possible acute beneficial effects that FOJ can exert in healthy humans by modulating postprandial glucose and lipid metabolism, and antioxidant and inflammatory response, compared with OJ.

## 2. Materials and Methods

### 2.1. Chemicals and Reagents

All Reagents and chemicals used in this study were purchased from Sigma-Aldrich Qúimica (Madrid, Spain).

### 2.2. Production of Fermented Orange Juice

FOJ was produced by Grupo Hespérides Biotech S.L. (Seville, Spain) and Mitra Sol Technologies S.L. (Alicante, Spain). A commercial OJ from *Citrus sinensis* L. var. Navel late was subjected to controlled alcoholic fermentation during ten days at 22 °C in 100 L stainless steel tanks. The yeast strain used was *Pichia kluyveri* var. kluyveri (previously isolated from the natural microbiota in orange fruit) which ferments only reducing sugars. The alcohol content of FOJ was low (1% *v*/*v*). The FOJ obtained was subsequently subjected to a pasteurization treatment (80 °C for 30 s) in a semi-industrial tubular pasteurizer (Mipaser Prototype, Murcia, Spain) and cooled in an ice-water-bath (10 °C). The next processes were carbonation (0.44 × 105 Pa) and aseptic packaging in aluminum containers (250 mL) which were stored at 4 °C until their consumption. OJ and FOJ quality parameters were measured using the OIV procedures (2017) [58].

### 2.3. Subjects

Nine healthy subjects (seven female and two male) aged 21–25 years were recruited for the study from the student population of Pablo de Olavide University (Seville, Spain). The selection process was based on haematological and biochemical parameters, medical history, anthropometric values and a nutrition and lifestyle questionnaire. Exclusion criteria included: (1) the presence of chronic diseases, overweight, and/or kidney or liver failure; (2) abnormal biochemical values; (3) consumption of any medication or dietary supplement during the previous 4 weeks; (4) any smoking habit; and (5) alcohol consumption of >2 drinks per day. All subjects gave their oral and written informed consent to participate in the study, which followed the principles of the Declaration of Helsinki. The study was approved by the Clinical Research Ethical Committees of Virgen del Rocío Hospital (IEC 2013PI/022; Seville, Spain) and Pablo de Olavide University (Seville, Spain).

### 2.4. Experimental Design

A randomized, controlled, and cross-over trial was conducted (Figure 1). 

Volunteers were asked to avoid carotenoid an (poly)phenol-rich foods for 48 h previous and during the trial. Subjects arrived at the clinical laboratory (Sevilab S.L. Seville, Spain) after overnight fasting for 12 h. Following a baseline blood draw (0 h), 500 mL of OJ was consumed at 8:00 h within 15 min (OJ treatment). Consecutive blood samples were collected at 1, 2, 3, 4, 5, 6, and 8 h. After blood draw at 4 h, subjects consumed a standardized mixed meal low in carotenoids and (poly)phenols composed of a sandwich with two slices of bread (60 g), cooked turkey (40 g) and slice of cheese (20 g), and a yogurt (125 g). The nutritional content of this meal of 245 kcal was 3.2 g total fat, 20 g total protein, and 34.2 g total carbohydrates. Only water (ad libitum) was allowed during the 8 h blood collection time. Following a two-week wash-out period, the intervention was carried out with the acute intake of FOJ (FOJ treatment). 

### 2.5. Blood Sample Collection and Handling

Blood samples (18 mL) were drawn from a forearm vein into EDTA tubes and were centrifuged (3500 rpm for 3 min at 4 °C) to obtain plasma and erythrocytes samples, which were divided into 0.5 mL aliquots and stored at −80 °C until analysis. Biochemical meaurements (Glucose, creatinine, glutamic oxaloacetic transaminase (GOT), glutamic-pyruvic transaminase (GPT), hemoglobin, red blood cell levels, triglycerides (TG), total cholesterol (TC), HDL cholesterol (HDL-C) and LDL cholesterol (LDL-C) were carried out by the clinical laboratory of Sevilab.

### 2.6. Anthropometric Evaluation

Weight, height and waist circumference (WC) were obtained while volunteers were in a fasting state, barefoot and wearing only non-restrictive undergarments. A model 780 SECA digital scale balance (SECA, Barcelona, Spain) was used to measure body weight and height, from which the body mass index (BMI) (kg/m^2^) was calculated. WC was measured around the narrowest part of the torso using a model 201 SECA flexible tape. All measurements were taken using standardized protocols [59].

### 2.7. Glucose Metabolism

Glucose levels were evaluated in plasma samples by enzymatic spectrophotometry (Roche Diagnostics, Madrid, Spain) using a Roche Cobas 8000 Modular Analyzer (Roche Diagnostics). The following parameters related to the postprandial glycemic response were analyzed: glucose baseline (GB) is the glucose level previous to meal intake; peak glucose value (PGV) is the highest value observed after ingesting the meal (mg/dL); absolute increase in glucose (AIg) is the absolute difference between PGV and GB (mg/dL); glucose incremental percentage (GIP) is the ratio between AIg and GB (%); glucose incremental velocity (GIV) is the ratio between AIg and the time in minutes where the peak value was recorded (mg/mL minute).

### 2.8. Lipid Metabolism

Levels of TC, LDL-C, HDL-C and TG were analyzed in plasma samples by enzymatic spectrophotometry (Roche Diagnostics, Madrid, Spain) using a Roche Cobas 8000 Modular Analyzer (Roche Diagnostics). Plasma apolipoprotein (apo)-AI and apo-B concentrations were performed by enzymatic methods using Atellica CH 930 Analyzer (Siemens Healthcare Diagnostics, Marburg, Germany).

### 2.9. Antioxidant and Inflammation Status

Antioxidant status was estimated through plasma antioxidant capacity (PAC) and endogenous antioxidant compounds.

PAC was assessed by the assays Oxygen Radical Absorbance Capacity (ORAC) [60] and Ferric Reducing Antioxidant Power (FRAP) [61]. The plasma samples were diluted (1:1000) in phosphate buffer (75 mM, pH 7.4) and (1:8) in distilled water, respectively. A Synergy™ HT-multimode microplate reader (Biotek Instruments, Winooski, VT, USA) was used to obtain PAC values. 

Levels of the endogenous antioxidants albumin, bilirubin and uric acid were evaluated in plasma samples by enzymatic spectrophotometry using a Roche Cobas C 701 modular analyzer (Roche Diagnostics Systems Inc., Branchburg, NJ, USA).

Plasmatic C-reactive protein (CRP) and fibrinogen content were analyzed using a Roche Cobas c7002 modular analyzer (Roche Diagnostics Systems Inc.) and Stago Compact Max (Barcelona, Spain), respectively. 

### 2.10. Statistical Analysis

Data are expressed as mean ± standard error of the mean (SEM). The Shapiro–Wilk test was applied to assess the normal distribution of the values. Comparisons of the different parameters in each group for each time-point, and between OJ and FOJ groups, were assessed using the Two-way analysis of variance (ANOVA) followed by Bonferroni post hoc test. The area under the curve (AUC) values were compared in each group using Student’s *t*-test on normally distributed variables and the Mann–Whitney U test when they were not. A probability value of *p* < 0.05 was adopted as the criteria for significant differences. All statistical analyses were carried out by using GraphPad Prism v8 software (San Diego, CA, USA).

## 3. Results and Discussion

This study aimed to determine the postprandial effect of FOJ versus OJ on glucose and lipid metabolism and antioxidant/inflammatory response in healthy humans.

The composition (quality parameters, nutritional profile, bioactive compounds content and antioxidant activity) of OJ and FOJ are shown in Table 1. 

The percentage of pulp in FOJ was lower than in OJ, with its consequently reduced soluble fiber content. The total sugars were lesser in FOJ compared with OJ since the yeast strain ferments reducing sugars. The FOJ can be considered a rich source of bioactive compounds incluiding flavanones (667.2 μmol/L), carotenoids (10.33 μg/mL), provitamin A (70.97 μg/L) and melatonin (0.192 ng/mL), which were described previously [40,41].

The baseline characteristics of the subjects included in the study are shown in Table 2. 

The values were within the desired ranges and did not significantly differ between the two intervention periods (data not shown). All participants reported adhering to dietary restriction and complied with all evaluation procedures. Both beverages (OJ and FOJ) were well tolerated by all volunteers. No adverse events were reported.

### 3.1. Effect of Acute FOJ Consumption on Postprandial Glucose Metabolism

Figure 2 shows glucose levels evolution throughout the entire experiment (0–8 h) after OJ or FOJ consumption. 

Although the total sugar content of FOJ (47.9 g/L) is approximately half that of OJ (78.2 g/L) (Table 1), blood glucose followed a similar trend before meal intake (0–4 h). Decrease from time 0 until 1 h after the single intake (OJ: ↓ 19.2%; FOJ: ↓ 20.3%), and slight increases at 2 h and 3 h, with values similar to baseline at 4 h. The values are significantly different between times 0 and 2 h in both beverages (OJ: *p* = 0.01; FOJ: *p* = 0.01), and between 0 and 3 h in OJ (*p* = 0.04). Our results are similar to other authors. Bosch-Sierra et al. (2019) [55] compared two types of orange juice with different fiber content, and both showed a decrease in blood glucose after 1 h of ingestion compared with baseline, and a slight increase at 2 h. Since no intermediate times were measurement between 0 and 1 h, the increase magnitude of a single intake of OJ vs. FOJ on plasmatic glucose could not be evaluated. Previous studies obtained a maximum increase in glucose levels at 15–30 min after consumption of OJ and values equal to or less than baseline at 60 min [55,56]. Other authors used time intervals greater than to that of current study (2–3 h) [48,49]. 

After ingestion of the meal at 4 h, the evolution of both beverages was also not statistically different: Increase in plasmatic glucose from 4 to 5 h (0–1 post meal intake), and progressive decrease from 5 to 8 h (1–4 h post meal intake) (when the values return to pre-intake values). Table 3 summarizes the mean AUC values of all markers.

There was no statistical difference in glucose AUC values between both treatments. The current results differ of other authors. Cerletti et al., (2015) [49] measured blood glucose levels at 2 h after intake of red or blond OJ with a fatty meal. With both juices, blood glucose decreased significantly from 96 to 85 mg/dL. In our study, up to 4 h the values are not similar to those prior to the intake of meal. The discrepant results between studies could be due to differences in meal composition and experimental design.

The values of the glucose parameters (GB, PGV, AIg, GIP, HP, GIV) in the postprandial curves (times 4–8 h/1–4 h post meal intake) are shown in Table 4. 

No significant differences were obtained between the two beverages for any parameter. However, the glycemic response of meal was attenuated with FOJ intake compared with OJ: PGV, AIg and GIP values were lower after FOJ consumption than those after OJ. In addition, the peak blood glucose (HP) was maintained longer in the OJ curve than in that of FOJ, and the GIV or glycemic index was higher with OJ than with FOJ. The lack of significance could be because of the small sample size of this study. 

Flavonoids have important functions in gastrointestinal tract, i.e., glucose homeostasis via slow digestion of carbohydrates and intestinal glucose absorption [62,63]. Various mechanisms have been proposed for these effects, including combination of noncovalent interactions with α-amylase and α-glucosidases, and interactions that alter expression and efficiency of intestinal glucose transporters [56,64]. Such effects have been shown with the orange-derived flavanone naringenin [65,66] and hesperetin [67]. In a previous study, absorption, metabolism, and excretion of OJ versus FOJ (poly)phenols in humans were evaluated [40]. Plasma levels of naringenin and hesperetin derivatives were higher after ingestion of FOJ in relation to OJ, reaching maximum values between 3 and 5 h after intake. On the other hand, noncovalent interactions between flavonoids and soluble fiber limit their availability to interact with transporters at the intestinal brush border [64]. The reduced soluble fiber content in FOJ due to the fermentation process could also enhance of hypoglycemic effect in relation to OJ.

### 3.2. Effect of Acute FOJ Consumption on Postprandial Lipid Metabolism

Figure 3 shows the lipid profile (TC, LDL-C, HDL-C, TG, apo-A1 and apo-B values) in plasma samples throughout the entire experiment (0–8 h) after OJ or FOJ consumption.

TC, LDL-C, HDL-C, TG and apo-A1 did not vary statistically over time (0–8 h) or between beverages (OJ and FOJ) (Figure 3A–E). The levels of these markers were unaffected after meal intake (times 4–8 h/1–4 h post meal intake). Additionally, the AUC values were not significantly different among the treatments for any parameter (Table 3). However, a non-significant tendency towards increases in LDL-C values were shown after 1 h of consumption in comparison with basal level, higher with OJ (↑ 39.1%) that the value obtained with FOJ (↑ 23.8%) (Figure 3B). Several studies with OJ also found no significant effects on lipid profile. TC, LDL-C and HDL-C did not differ significantly at 3 h after intake of OJ with a breakfast compared with drinking water in adult humans [48]. Coelho et al. (2017) [68] also found no changes in TC, LDL-C and HDL-C levels after acute consumption (0–5 h) of OJ with high-fat meal versus water. Cerletti et al. (2015) [49] did not obtain significant difference in cholesterol level between OJ and water 2 h post a fatty meal consumption.

A significant decrease in apo-B concentration was found 4 h (*p* = 0.03) after consumption of FOJ, compared with baseline value, but no significant time effect was found for the OJ treatment (Figure 3F). The level of apo-B was unaffected after meal intake (times 4–8 h/1–4 h post meal intake), and the AUC values were not significantly different among the treatments for any parameter (Table 3).

Apo-B has been shown to be superior to LDL-C in predicting the risk of vascular events and the progression of vascular disease. Lipid concentration of LDL-C fraction does not necessarily equal the total number of LDL particles and there is not a precise relationship between both markers [69]. Regarding the fact that each LDL-C particle contains one molecule of apo-B and more than 90% of total plasma apo-B is associated with LDL-C particles [70], the reduction in apo-B in the FOJ treatment could indicate that the number of LDL-C particles decreased thus lowering the atherogenic risk. In other acute intake studies of juices and fruits no significant changes in apo-B were obtained. So, there were no effect in serum apo-B after elderberry juice or blueberries ingested simultaneously with high-fat and/or high-sugar meal [71,72].

TG levels followed a similar trend before meal intake (0–4 h) in both curves (OJ and FOJ): A decrease from 0 until 2 h (significant in OJ curve, *p* = 0.04), and maintenance between 2 and 4 h. It should also be noted that, after eating meal at 4 h, there is a maximum increase in TG of ↑ 23.1% in the OJ curve and of ↑ 7.4% in the FOJ group. Thus, a trend of a smaller increase in postprandial triglyceride levels after the intake of FOJ was evaluated. The lack of significance could be because of the small sample size of this study. In previous studies OJ intake also attenuated the increase in postprandial TG levels promoted by high-fat meals compared with water control [48,49,68]. The results were similar using other fruit juices (Pear juice [54]; Elderberry juice [71]).

In addition to the low content of fat in the food (3.2 g total fat), the triglyceride levels would have also possibly been increased by the fructose content of OJ and FOJ, as has been previously reported [73,74]. The reduction in sugar content produced by the alcoholic fermentation process means that FOJ may be healthier than the original OJ in relation to the maintenance of TG levels, in addition to the potential lipid-lowering effect of flavonoid metabolites, increased in plasma after consumption of FOJ versus OJ [40]: Plasma levels of hesperetin derivatives and naringenin derivatives were higher after ingestion of FOJ in relation to OJ, reaching maximum values between 3 and 5 h after intake. The beneficial role of bioactive compounds in moderate response of plasmatic TG has been evaluated [75].

Postprandial TG levels are known as risk predictors of coronary heart disease [76,77], and delayed clearance of TG-rich lipoproteins (postprandial hyperlipidemia) is associated with the development of atherosclerosis [78]. Therefore, improvement of the vascular prognosis might be achieved through the reduction in plasmatic TG in the postprandial phase. FOJ intake could reduce the increase in postprandial TG levels in relation to OJ and play a protective role against this cardiovascular risk factor. 

### 3.3. Effect of Acute FOJ Consumption on Antioxidant and Inflammation Status

Figure 4 shows PAC measured by ORAC and FRAP assays throughout the entire experiment (0–8 h) after OJ or FOJ consumption.

The evolution of the ORAC values was different between the two treatments (Figure 4A). In the case of FOJ, a notable increase from the basal value to the value at 2 h was obtained (↑ 29.7%). Subsequently, values declined until 4 h, and increased slightly until 8 h. In the case of OJ, there are fluctuations over time, but at no time ORAC values were higher than baseline. Although significant differences between OJ and FOJ were not observed at any time after ingestion, the AUC of ORAC values after FOJ intake was significantly higher in the FOJ group compared with OJ intake (*p* = 0.04) (Table 3). 

A significant increase in ORAC values was also observed after consumption of jabuticaba juice, peaking at 90 min, and the postprandial AUC was higher compared with the intake of the control beverage [50]. 

The improvement of the antioxidant status after the consumption of FOJ in relation to OJ, revealed by a higher AUC value, was probably due to the presence of intestinally absorbed polyphenols and/or their derived metabolites in the blood of the test subjects. Plasma levels of naringenin derivatives, phenolic acids and hesperetin derivatives were higher after ingestion of FOJ in relation to OJ up to 6 h post ingestion [40]. In addition, phenolic acids and naringenin derivatives presented the highest values between times 1 and 3 h, and a decrease at 4 h, coinciding with the decrease in ORAC value described after ingestion of FOJ.

No significant changes were observed in FRAP values over time (0–8 h) or between beverages (Figure 4B), in agreement with previous studies. No effect was observed on plasma FRAP after intake of grape or pomegranate pomace dietary supplement [79]. 

The selected antioxidant activity assays are based on different chemical reactions, solvents and radical species. Thus, antioxidant capacity results may differ among the measurement methods [80]. The PAC values were unaffected after meal intake (times 4–8 h/1–4 h post meal intake).

Plasma has various endogenous antioxidant molecules. Albumin, bilirubin and uric acid are the major antioxidant components of plasma and contributors to its antioxidant capacity [81,82]. Figure 5 shows endogenous antioxidants content (albumin, bilirubin and uric acid) after OJ or FOJ consumption.

Albumin levels were significantly increased in the OJ group at 1h post-ingestion with respect to the baseline (*p* = 0.04) (Figure 5A). FOJ intake did not present significant changes in albumin levels. Bilirubin values underwent a significant increase in FOJ at 3 h (*p* = 0.03) and 4 h (*p* = 0.002) post-intake, compared with the basal values (Figure 5B). A similar pattern was observed in the OJ group (4 h, *p* = 0.001; 5 h, *p* = 0.01; 6 h, *p* = 0.04). The albumin and bilirubin levels were unaffected after meal intake (times 4–8 h/1–4 h post meal intake). No significant differences were found at any sampling time among the treatments (OJ vs. FOJ) in albumin, bilirubin and uric acid levels. Similarly, there was no statistical difference in AUC values in either parameter (Table 3).

In relation to uric acid, no significant increase was registered at 1 h post intake of OJ or FOJ in relation to baseline value. After this time, levels were significantly decreased at 2 h post-ingestion of FOJ compared to basal values (*p* = 0.02), and after eating the meal (between 4–5 h), other increase was obtained (Figure 5C). In the OJ group significant lower values at 2 h (*p* = 0.003) also were obtained, but the levels decreased until the end of the intervention (8 h), being unaffected after meal intake (times 4–8 h/1–4 h post meal intake). These results were similar those obtained by other authors. Compared with baseline values, blood uric acid levels were significantly increased after the single consumption of Bordo grape wine at 1 h post intake. However, uric acid decreased after the consumption of Bordo grape juice [45].

Fructose contained in both beverages could intervene in the uric acid enhancements described [83]. Moreover, the flavonoids of FOJ could increase plasma uric acid after meal consumption, according to Lotito and Frei (2006) [83]. The increase in antioxidant activity (ORAC value) described could be mainly caused by flavonoid metabolites and not by urate since that uric acid particularly contributes to the antioxidant capacity of serum assessed by the FRAP assay [84,85].

Figure 6 shows CRP and fibrinogen levels in plasma samples throughout the entire experiment (0–8 h) after OJ or FOJ consumption.

No significant time effect was found for fibrinogen values in the plasma when compared between time points within each treatment (Figure 6A). Additionally, the fibrinogen plasma concentration was not significantly different among the treatments at any point. A significant decrease in CRP level was found at 3 h (*p* = 0.03) and 4 h (*p* = 0.04) after consumption of OJ, compared with baseline value. No significant time effect was found for the FOJ treatment (Figure 6B). The high SEM values obtained in the CRP measurements in the FOJ treatment could explain the lack of effect. There was no statistical difference in CRP and fibrinogen AUC values among both treatments (Table 3).

Although both the intake OJ and FOJ increased plasma levels of naringenin and hesperetin derivatives from 3 h of intake [40], the anti-inflammatory effect of OJ could also be due to its high level of vitamin C (423 mg/L). The anti-inflammatory effect of vitamin C has been extensively evidenced previously [86].

Anti-inflammatory evaluation of juices linked to acute intake or postprandial stress is limited. In healthy subjects, the concomitant intake of OJ also decreased the low-grade inflammatory reaction induced by a fatty meal at 2 h post intake [49], although markers of cell activation (platelet p-selectin, leukocyte Mac-1 expression and myeloperoxidase (MPO) content) were used. Peluso et al. (2014) [87] showed that the consumption of fruit juice reduced IL-6 at 2 h and TNF-a at 1–8 h post intake in response to a high-fat meal.

## 4. Conclusions

Our data indicate that in healthy humans, the acute intake of fermented orange juice ameliorated plasma LDL-C, significantly decreased apo-B and increased plasma antioxidant capacity. This study also described a postprandial effect of fermented orange juice through the attenuation of plasma glucose and triglyceride levels compared with orange juice, probably due to its suitable profile of highly bioavailable (poly)phenols.

The regular consumption of a single doses of fermented orange juice could exert antioxidant and lipid-lowering effects and modulate the postprandial metabolic response. Thus, fermented orange juice could be a new functional food with potential positive effects on the maintenance of health status and prevention of chronic diseases.

Further studies are needed to confirm these results and to elucidate the role of the bioactive components involved in these effects.

## Figures and Tables

**Figure 1 foods-11-01256-f001:**
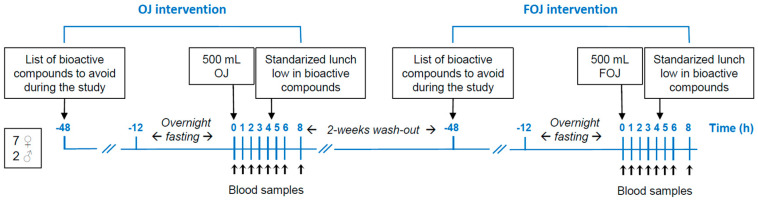
Flow diagram of the progress through the phases of the randomized, controlled, and cross-over intervention.

**Figure 2 foods-11-01256-f002:**
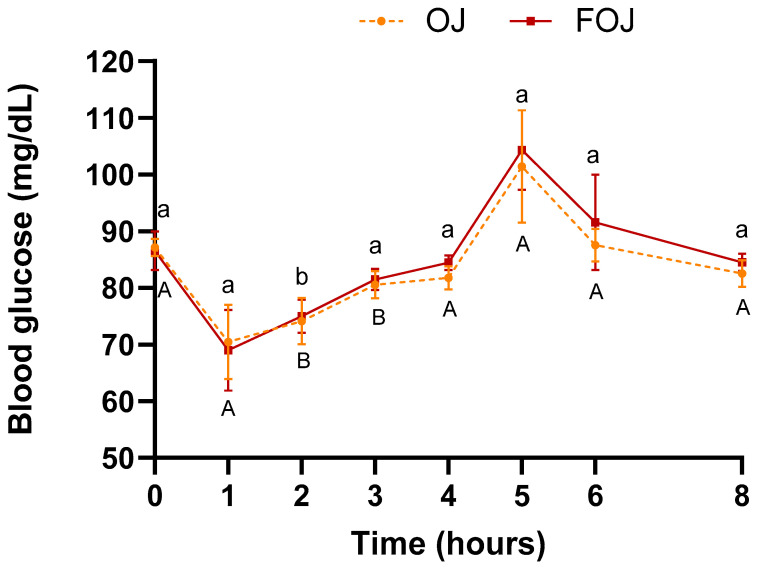
Glucose levels throughout the entire experiment (0–8 h) after orange juice (OJ) or fermented orange juice (FOJ) consumption. Data points represent the mean ± SEM (*n* = 9). Not significant difference (*p* < 0.05) between OJ and FOJ treatments. Different capital and lowercase letters indicate the statistical difference (*p* < 0.05) within the OJ and FOJ groups at the different time points, respectively.

**Figure 3 foods-11-01256-f003:**
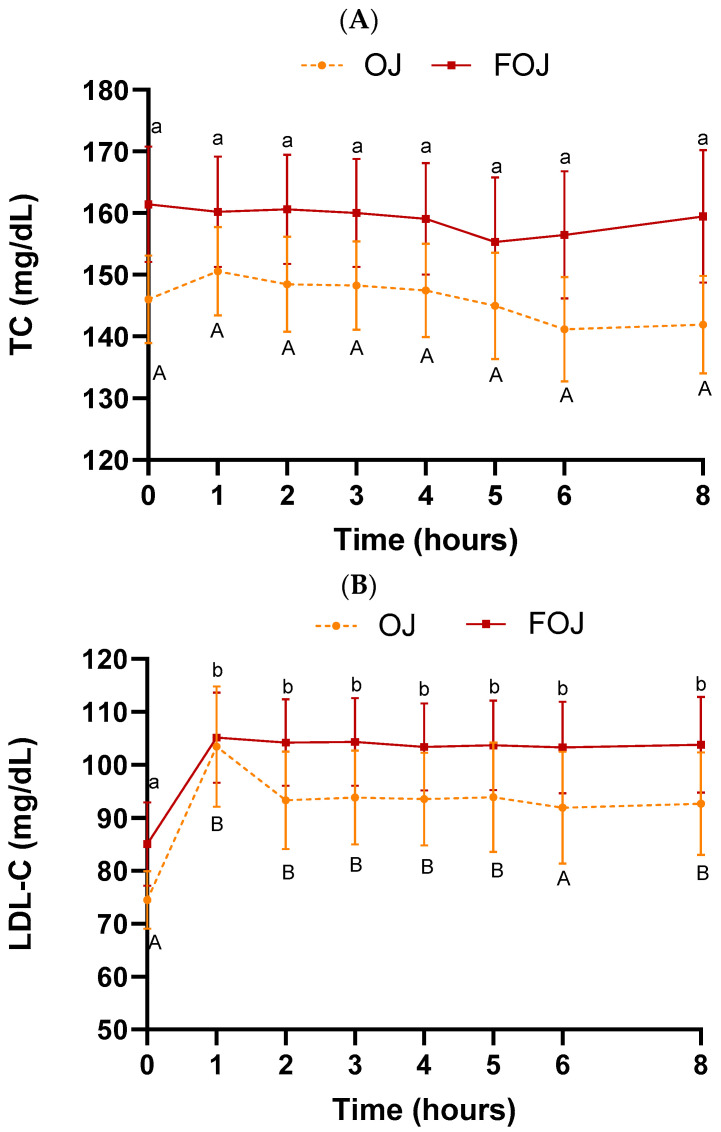
Lipid metabolism in plasma samples throughout the entire experiment (0–8 h) after orange juice (OJ) or fermented orange juice (FOJ) consumption. (**A**) Total cholesterol (TC), (**B**) LDL cholesterol (LDL-C), (**C**) HDL cholesterol (HDL-C), (**D**) Triglycerides (TG) values (mg/dL), (**E**) Apolipoprotein (apo)-A1 and (**F**) apo-B values (mg/dL). Data points represent the mean (*n* = 9) ± SEM. Not significant difference (*p* < 0.05) between OJ and FOJ treatments. Different capital and lowercase letters indicate the statistical difference (*p* < 0.05) within the OJ and FOJ groups at the different time points, respectively.

**Figure 4 foods-11-01256-f004:**
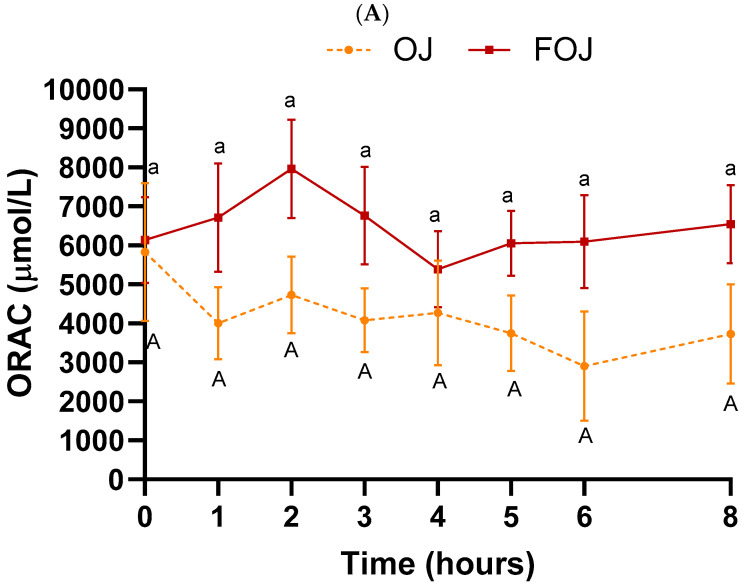
Plasma antioxidant capacity (PAC) throughout the entire experiment (0–8 h) after orange juice (OJ) or fermented orange juice (FOJ) consumption. (**A**) Oxygen Radical Absorbance Capacity (ORAC) assay. (**B**) Ferric Reducing Antioxidant Power (FRAP) assay. Data points represent the mean (*n* = 9) ± SEM. Same capital and lowercase letters indicate not significant difference (*p* < 0.05) between OJ and FOJ treatments. Not significant difference (*p* < 0.05) within the OJ and FOJ groups at the different time points, respectively.

**Figure 5 foods-11-01256-f005:**
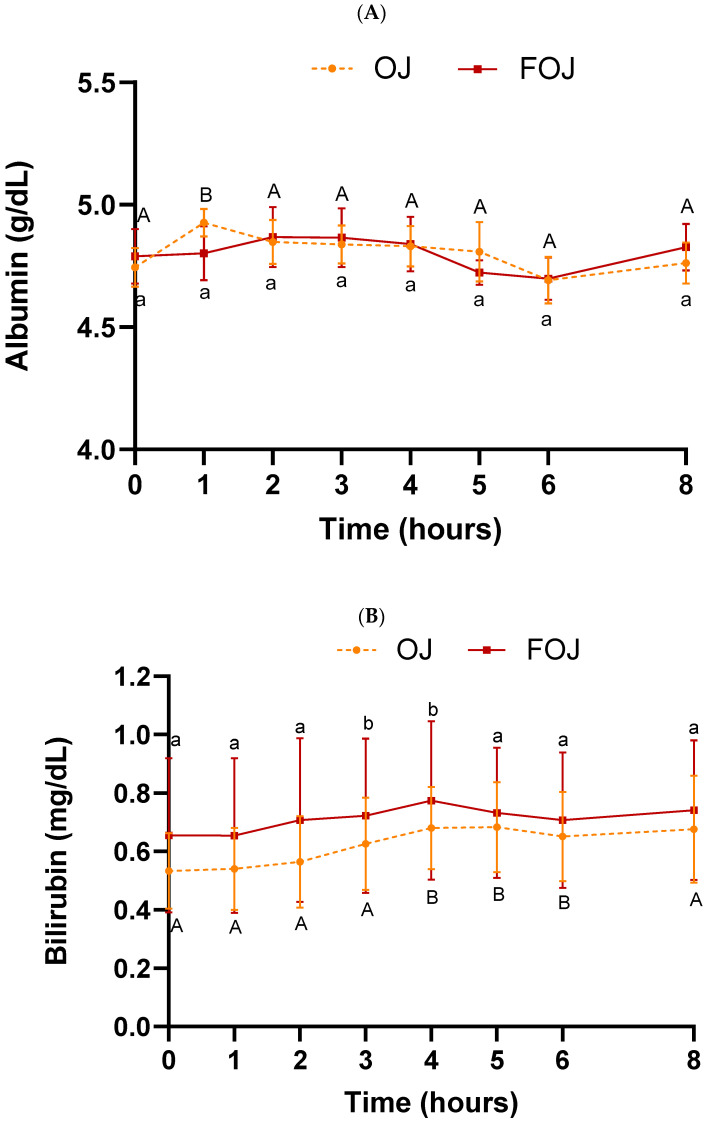
Endogenous antioxidants content in plasma samples throughout the entire experiment (0–8 h) after orange juice (OJ) or fermented orange juice (FOJ) consumption. (**A**) Albumin (g/dL), (**B**) Bilirubin (mg/dL) and (**C**) Uric acid (mg/dL) values. Data points represent the mean (*n* = 9) ± SEM. Not significant difference (*p* < 0.05) between OJ and FOJ treatments. Different capital and lowercase letters indicate the statistical difference (*p* < 0.05) within the OJ and FOJ groups at the different time points, respectively.

**Figure 6 foods-11-01256-f006:**
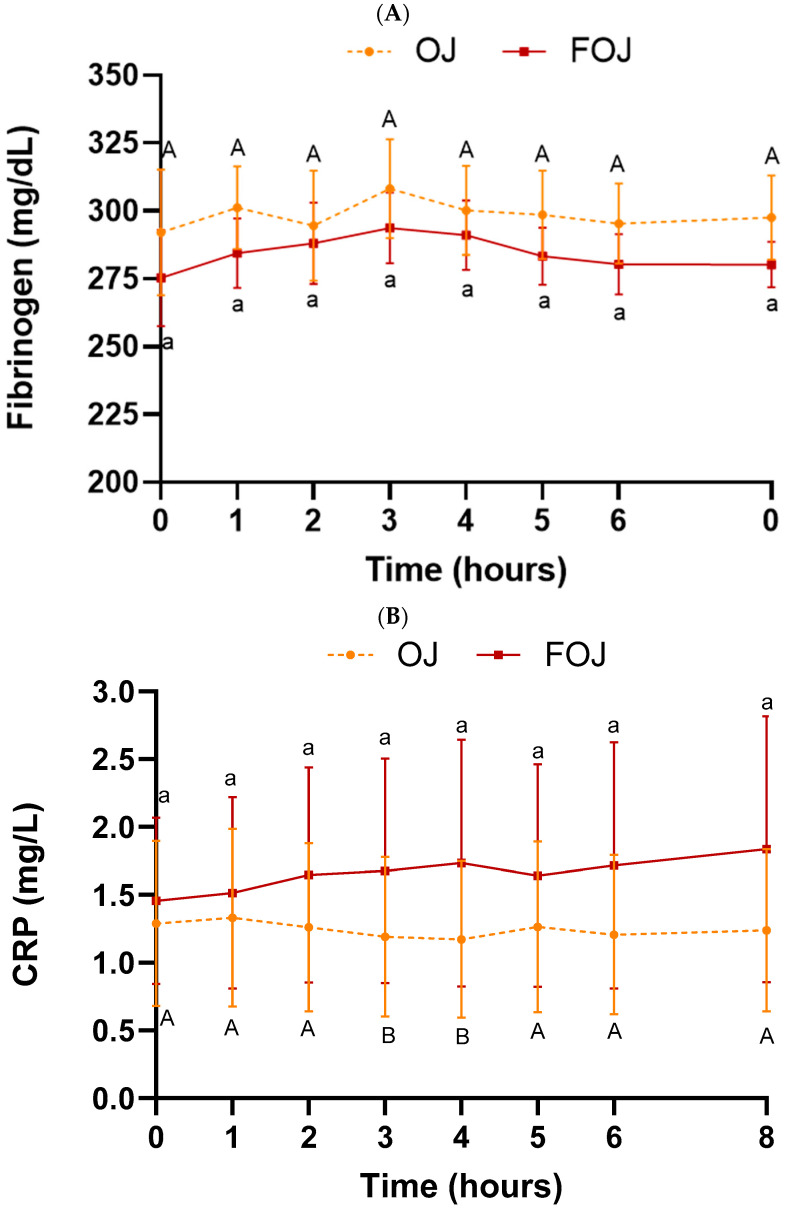
Inflammation status in plasma samples throughout the entire experiment (0–8 h) after orange juice (OJ) or fermented orange juice (FOJ) consumption. (**A**) Fibrinogen (mg/dL) and (**B**) C-reactive protein (CRP) (mg/L) values. Data points represent the mean (*n* = 9) ± SEM. Not significant difference (*p* < 0.05) between OJ and FOJ treatments. Different capital and lowercase letters indicate the statistical difference (*p* < 0.05) within the OJ and FOJ groups at the different time points, respectively.

**Table 1 foods-11-01256-t001:** Composition (quality parameters, nutritional profile, bioactive compounds content and antioxidant activity) of orange Juice (OJ) and fermented orange juice (FOJ).

Composition	OJ	FOJ
pH	3.48 ± 0.20	3.48 ± 0.03
TA (g citric acid/L)	8.48 ± 0.02	8.60 ± 0.01
TSS (ºBrix)	11.00 ± 0.50	9.01 ± 0.14
% Pulp	12.00 ± 2.00	5.65 ± 0.22
Alcohol (% *v*/*v*)	0	0.90 ± 0.15
Ascorbic acid (mg/L)	423.00 ± 1.80	197.00 ± 6.80
Total sugars (g/L)	78.20 ± 5.64	47.90 ± 4.10
Reducing sugars (g/L)	48.50 ± 3.63	20.30 ± 2.40
Non reducing sugars (g/L)	29.70 ± 2.01	27.10 ± 2.61
Total (poly)phenols (µmol/L)	659.0 ± 10.5	835.7 ± 37.6
Total flavanones (µmol/L)	502.7 ± 20.7	667.2 ± 31.6
Total flavones (µmol/L)	132.8 ± 6.0	142.8 ± 6.1
Total phenolic acids (µmol/L)	23.5 ± 4.2	25.7 ± 0.1
Total individual carotenoids (µg/mL)	11.04 ± 0.20	10.33 ± 0.40
Provitamin A RAEs (µg/L)	62.93 ± 1.85	70.97 ± 3.26
Melatonin (ng/mL)	0.185 ± 0.033	0.192 ± 0.072
ORAC (μmol/L)	6361 ± 261	6353 ± 307

Values are expressed as means ± SD (*n* = 3). TA, titratable acidity; TSS, total soluble solids; RAEs, Retinol Activity Equivalents; ORAC, oxygen radical absorbance capacity.

**Table 2 foods-11-01256-t002:** Anthropometric characteristics and biochemical measurements of subjects at baseline.

Characteristics	Means ± SEM
Age	22.00 ± 0.44
Body Weight (kg)	63.39 ± 2.60
Height (m^2^)	1.73 ± 0.08
BMI (kg/m^2^)	20.87 ± 0.49
WC (cm)	72.30 ± 1.22
Glucose (mg/dL)	86.83 ± 1.81
GOT (U/L)	18.72 ± 1.38
GPT (U/L)	15.17 ± 1.28
Creatinine (mg/dL)	0.80 ± 0.03
Hemoglobin (g/dL)	13.48 ± 0.30
Erythrocytes (×10^12^/L)	4.48 ± 0.10
Triglycerides (mg/dL)	56.83 ± 4.87
Total cholesterol (mg/dL)	153.72 ± 6.02
HDL cholesterol (mg/dL)	62.61 ± 3.39
LDL cholesterol (mg/dL)	79.72 ± 4.84

Values are expressed as means ± SEM (*n* = 18). BMI, body mass index; WC, waist circumference; GOT, glutamate oxaloacetate transaminase; GPT, glutamate pyruvate transaminase.

**Table 3 foods-11-01256-t003:** Area under the curve (AUC) values of lipids, glucose, antioxidant status and inflammation status markers after consumption of orange juice (OJ) and fermented orange juice (FOJ) by 9 subjects.

Parameter	OJ	FOJ	*p*-Value
ORAC (µmol/L)	26,951.0 ± 5798.2	44,093.3 ± 5361.6 *	0.04
FRAP (mmol/L)	16,606.1 ± 1749.5	14,986.2 ± 467.1	0.38
Albumin (g/dL)	32.1 ± 1.6	33.6 ± 0.6	0.98
Bilirubin (mg/dL)	4.1 ± 1.0	5.0 ± 1.8	0.81
Uric Acid (mg/dL)	26.3 ± 1.8	27.8 ± 1.7	0.53
Glucose (mg/dL)	550.2 ± 33.0	591.4 ± 17.0	0.66
TC (mg/dL)	976.3 ± 67.9	1112.4 ± 65.7	0.16
LDL-C (mg/dL	621.9 ± 64.2	718.6 ± 58.3	0.28
HDL-C (mg/dL)	410.1 ± 41.9	458.6 ± 31.0	0.36
TG (mg/dL)	380.1 ± 56.1	375.0 ± 20.8	0.93
TC/HDL-C	16.4 ± 1.4	17.3 ± 1.1	0.63
LDL-C/HDL-C	10.7 ± 1.5	11.3 ± 1.1	0.74
Apo-AI (mg/dL)	794.3 ± 50.4	840.2 ± 26.6	0.43
Apo-B (mg/dL)	366.6 ± 29.3	409.7 ± 23.1	0.26
C-reactive protein (CRP) (mg/L)	8.3 ± 4.1	11.6 ± 5.8	0.43
Fibrinogen (mg/dL)	1997.9 ± 165.3	1998.7 ± 79.2	0.99

Values are expressed as means ± SEM (*n* = 9). ORAC, oxygen radical absorbance capacity; FRAP, ferric reducing antioxidant power; TC, total cholesterol; LDL-C, low-density lipoprotein cholesterol; HDL-C, high-density lipoprotein cholesterol; TG, triglycerides; Apo, apolipoprotein; CRP, C-reactive protein. * Significant at *p* < 0.05 in the mean AUC between OJ and FOJ groups.

**Table 4 foods-11-01256-t004:** Effects of consumption of orange juice (OJ) or fermented orange juice (FOJ) on postprandial glycemia in all subjects.

Parameter	OJ	FOJ	*p* Value *
GB (mg/dL)	81.0 ± 2.2	84.4 ± 1.3	0.10
PGV (mg/dL)	107.5 ± 7.7	106.4 ± 7.4	0.92
AIg (mg/dL)	26.5 ± 8.7	22.0 ± 7.9	0.70
GIP (%)	33.9 ± 11.0	26.6 ± 9.9	0.63
HP (min)	360 ± 27.8	333.3 ± 20.3	0.57
GIV (%)	8.6 ± 3.0	6.8 ± 2.3	0.64

Values are expressed as means ± SEM (*n* = 9). GB, basal glucose; PGV, Maximum glucose value; AIg, maximum glucose increase; GIP, incremental glucose percentage; HP, peak blood glucose time; GIV, incremental glucose rate. * Significant at *p* < 0.05 in the mean values between OJ and FOJ groups.

## Data Availability

The data presented in this study are available on request from the corresponding author.

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
