# Peer review of "Effect of Acute Intake of Fermented Orange Juice on Fasting and Postprandial Glucose Metabolism, Plasma Lipids and Antioxidant Status in Healthy Human"

_foods, 2022, doi:10.3390/foods11091256_

Round 1

Reviewer 1 Report

 The authors aimed to evaluate the postprandial response of the acute ingestion of fermented orange juice (FOJ) compared to orange juice (OJ) in humans in relation to inflammatory/antioxidant status in a crossover study. The methods were appropriated to answer the research question. However, I felt a lack of data from a control group, without juice intake, for comparison purposes.

The authors assumed that the most important result was that the glycemic and triglyceride levels were attenuated by FOJ. But no significant differences were obtained between the two beverages for any glycemic parameters (table 4). Another concern is about the affirmation at pg. 12 lines 361-363: “Thus, the AUC value is higher, although not significantly, in the case of OJ compared to FOJ (Table 3)”. If the values had no significance, they are the same.

There was a trend of a smaller increase in LDL-C after FOJ intake compared to OJ but a significant decrease in apo-B, how the authors explain this controversial effect?

Probably the lack of significance among lipids results could be because of the meal offered was non-high fat, which reduces the postprandial effect of the meal and consequently the range for the effects of the juices.

General observations:

In the Abstract: In respect of antioxidant evaluation, there was a significant increase in ORAC, but the authors conclude that are no differences in endogenous antioxidants status between the treatments.

The introduction is too long, it can be summarized.

Table 3: include * for significant different values (ORAC)

Figures legend: there is no * in the figures, why do the authors include the phrase: * Significant difference (p<0.05) between OJ and FOJ treatments.

Orthographic corrections:

Pg. 1 Line 14:  potentail acute

Pg. 18 Line 510: the postpandrial

Reviewer 2 Report

General appraisal:

The study is interesting because it can lead to the production of functional food that helps combat metabolic disorders that occur in people at risk of cardiovascular disease and other chronic diseases.

Examining the text, it can be seen that: 

The title reflects the content of the article.

The abstract makes an adequate synthesis of all the work

The keywords are informative

The communication of ideas is clear and has an adequate organization of ideas.  Good spelling and grammar. It can be seen proper use and correct citation and referencing of tables and figures. The style throughout the document is consistent. Graphs and tables are readable.  

The interpretations and conclusions are justified by the data presented.

The presentation and organization of the article are adequate

The use of graphs and tables for presenting and interpreting the results is adequate

The references are up-to-date and the conclusions correspond to the work. 

However, the article could be improved.  I consider that the length of the article is too extensive. This is because the authors put in the text the results that are already explicit in the tables and graphs, which leads to unnecessary repetition.
